# An Update of the Promise of Glycine Supplementation for Enhancing Physical Performance and Recovery

**DOI:** 10.3390/sports12100265

**Published:** 2024-09-25

**Authors:** Arnulfo Ramos-Jiménez, Rosa Patricia Hernández-Torres, David Alfredo Hernández-Ontiveros, Melinna Ortiz-Ortiz, Reymond Josué López-Fregoso, José Miguel Martínez-Sanz, Genaro Rodríguez-Uribe, Marco Antonio Hernández-Lepe

**Affiliations:** 1Conahcyt National Laboratory of Body Composition and Energetic Metabolism (LaNCoCoME), Tijuana 22390, Mexico; aramos@uacj.mx (A.R.-J.); rhernant@uach.mx (R.P.H.-T.); david.hernandez@uabc.edu.mx (D.A.H.-O.); melinna.ortiz@uabc.edu.mx (M.O.-O.); reymond.lopez@uabc.edu.mx (R.J.L.-F.); josemiguel.ms@ua.es (J.M.M.-S.); genaro.rodriguez@uabc.edu.mx (G.R.-U.); 2Department of Health Sciences, Biomedical Sciences Institute, Autonomous University of Ciudad Juarez, Chihuahua 32310, Mexico; 3Faculty of Physical Culture Sciences, Autonomous University of Chihuahua, Chihuahua 31000, Mexico; 4Medical and Psychology School, Autonomous University of Baja California, Tijuana 22390, Mexico; 5Nursing Department, Faculty of Health Sciences, University of Alicante, San Vicente del Raspeig, 03690 Alicante, Spain; 6Academic Body “Salud Personalizada (UABC-CA-336)”, Autonomous University of Baja California, Tijuana 22390, Mexico

**Keywords:** human metabolism, physical exercise, sports medicine, athletic performance, oxidative stress, muscle function

## Abstract

Glycine, the simple amino acid, is a key component of muscle metabolism with proven cytoprotective effects and hypothetical benefits as a therapeutic nutrient. Cell, in vitro, and animal studies suggest that glycine enhances protection against muscle wasting by activating anabolic pathways and inhibiting proteolytic gene expression. Some evidence indicates that glycine supplementation may enhance peak power output, reduce lactic acid accumulation during high-intensity exercise, and improve sleep quality and recovery. This literature review critically explores glycine’s potential as an ergogenic aid and its relevance to muscle regeneration, muscle strength, endurance exercise performance, and sleep quality. It also underscores key areas for future research. It is concluded that more randomized controlled clinical trials in humans are needed to confirm glycine’s potential as a dietary supplement to support muscle function, recovery, and overall athletic performance as an ergogenic aid and to establish nutritional recommendations for athletic performance. Also, it is essential to consider that high doses (>500 mg/kg of body mass) could induce cytotoxic effects and contribute to acute glutamate toxicity.

## 1. Introduction

Throughout history, there has been a continuous interest in discovering the perfect formula to enhance physical and psychological performance. This is because performance significantly impacts general well-being, quality of life, longevity, and athletic achievements [1,2]. Consequently, there is a pressing need to identify foods, treatments, and innovative programs that can elevate our physical and psychological capacities while ensuring they are safe for consumption. The studies suggest that nutraceuticals have potential health benefits, including positive effects on cardiovascular and immune system health, anti-inflammatory and anti-cancer properties, and roles in preventing infections and metabolic disorders [3]. The beneficial effects of nutraceuticals are attributed to active compounds like carotenoids, collagen hydrolysate, dietary fibers, polyunsaturated fatty acids, antioxidants, and probiotics [4]. However, many nutraceutical supplements are consumed without first passing health and nutritional safety checks, which can result in liver damage [5], kidney damage [6], and cardiovascular damage [7], among other health issues. In addition, it is reported that around 50% of Americans (87% among athletes) take some form of nutraceutical supplement [8] and collectively spend more than USD 32 billion annually on these products [9], risking harm to their finances and health. Despite glycine being widely used as a nutraceutical supplement among athletes [10,11] for its health-promising preclinical evidence [12,13,14], there are no statistics on its consumption or its effects on athletic performance, highlighting the need for further research to bridge the translational gap. Therefore, this review aims to summarize the reports on using glycine as a sports nutritional supplement, emphasizing the need for safe and healthy supplementation and randomized, clinical controlled trials on this supplement.

## 2. Glycine’s Background

Glycine (Figure 1a), also known as amino acetic acid, plays a fundamental role in the biosynthesis of various biomolecules, such as glutathione, creatine, and purine nucleotides, and constitutes an essential element in the structure of numerous proteins (e.g., collagen in human connective tissue, Figure 1b). Abundant in plasma, it accounts for 11.5% of total amino acids and 20% of nitrogen content in body proteins, contributing significantly to total protein. Physiological glycine levels in human blood plasma range between 0.17 and 0.33 mM [15]. Approximately 35% of bodily glycine originates from endogenous synthesis (~3 g/day), 87% of them from serine via glycine-hydroxymethyltransferase (GMHT) [16]. In young males, glycine flux averages 34–35 mg/kg/h under fed conditions, decreasing by half (~18 mg/kg/h) during the post-absorptive state [17]. 

Glycine is a critical component of animal and plant proteins. Foods rich in glycine of animal origin include pork, beef, poultry, fish, dairy products, and eggs. As for foods of plant origin, glycine can be found in pumpkin, peas, carrots, beets, eggplant, sweet potatoes, potatoes, legumes, seeds, mushrooms, whole grains, nuts, and fruits. 

The recommended dietary intake of glycine ranges between 1.5 and 3.0 g/day [17]; however, it has been mentioned that this amount may not be sufficient for all the glycine metabolic processes [16], especially in elderly subjects [18], under disease processes and high-stress conditions such as schizophrenia patients, and in highly competitive sports environments [19]. Hence, there is an alleged need to increase this supplement’s intake under the abovementioned conditions [8,9]. Nevertheless, regular glycine supplementation has not been studied to be safe for humans. Regarding this topic, a retrospective study mentions that glycine supplementation may be associated with the risk of stroke mortality [20]. Similarly, it has been mentioned that high doses of glycine supplied over a short time, as occur in clinical situations where glycine is used routinely, may be hazardous [21]. 

The maximum doses of glycine administered without observing altered states or side effects in response to its ingestion were 60 g (30 g twice a day) in patients with schizophrenia [22] and 30 g (10 g three times a day) in marathon runners [11], but with no effect of glycine treatments in either study. Glycine intake, alone or in combination with other nutrients, has been extensively studied for its potential use as an anti-inflammatory, immunomodulatory, and cytoprotective agent in primary cells, cell lines, animal models, and people under pathological conditions (e.g., obesity, diabetes, cancer, and cardiovascular disease) [12,13,23]. In addition, it prevents tissue injury, enhances antioxidative capacity, and promotes protein synthesis and wound healing [24]. Lower circulating glycine levels are observed in obesity, type 2 diabetes (T2DM), and non-alcoholic fatty liver disease [25,26], and glycine supplementation has shown potential benefits in improving metabolic health and reducing the progression of obesity-related disorders [27]. Given these properties, glycine could benefit physical performance, including recovery, endurance, and muscle growth, but the safe doses have not been studied. This opens an emerging area of investigation with effects relevant to athletes and individuals engaged in physical training.

## 3. Biochemical Metabolism of Glycine in Mammals

The structure of glycine is characterized by a central carbon atom linked to an amino and a carboxyl group [28]. With a molecular weight of 75.07 g/mol, glycine is the lightest amino acid, and its amino and carboxyl side chains give it amphipathic properties, allowing it to interact in hydrophilic and hydrophobic environments [28]. Glycine is synthesized in all animal cells from choline, hydroxyproline, serine, sarcosine, and glyoxylate during endogenous synthesis of L-carnitine, with the liver and kidneys as the main producing organs [29]. In addition, glycine is nested in one-carbon metabolism into the Glycine Cleavage System, this being the primary glycine synthesis and degradation pathway, converting glycine into ammonia and CO_2_ and generating one-carbon units as 5,10-methylenetetrahydrofolate [24,29]. 

Currently, two kinds of glycine transporters are described: proton-coupled amino acid transporters (PATs) and glycine transporters (GlyTs) [30,31]. PATs from the SLC36 family are primarily involved in the transport of small, neutral amino acids in a proton-dependent manner and play a vital role in nutrient absorption, particularly in the intestines and kidneys, where they facilitate the uptake of dietary amino acids. They also contribute to cellular metabolism by maintaining amino acid homeostasis in various tissues [32]. GlyTs are Na^+^/Cl^−^-dependent neurotransmitter transporters responsible for l-glycine reuptake into the central nervous system [30,31]. 

Several biochemical roles of glycine in human metabolism (e.g., cytoprotective, neurotransmission, antioxidant, immunomodulation, and anti-inflammatory) have been published elsewhere [15,33,34,35]. These studies suggest that glycine’s cytoprotective action is due to its stimulation of stress protein synthesis, activation of extracellular signal-regulated kinase 1 and 2 (ERK1/2) and Akt-mTOR-FOXO1 signaling pathways, inhibition of chloride influx, stabilization of plasma membranes, and assistance in reducing the free radical’s production by glutathione synthesis [36]. These mechanisms maintain membrane integrity and cellular homeostasis and prevent inflammation and cellular death, especially during ATP depletion, hypoxia, and tissue injury during high stress [37,38]. In addition, glycine increases membrane hyperpolarization at the membrane level, acting as an inhibitory neurotransmitter [39].

## 4. The Role of Glycine in Skeletal Muscle Metabolism

Glycine is vital for muscle protein synthesis, growth, and repair. It is a building block of proteins and is considered a proteinogenic amino acid that plays a multifaceted role in muscle function, significantly affecting skeletal muscle metabolism, neuromuscular action, and exercise performance (Table 1). Animal studies suggest that glycine administration may preserve muscle mass, reduce inflammation, and increase growth hormone levels [13]. 

Approximately 5% of amino acids in muscle contractile protein are glycine, and one-third in structural proteins, such as collagen and elastin, highlighting its importance in muscle connective tissue and the extracellular matrix [40]. As a protein precursor, glycine contributes to collagen synthesis [40]. Collagen is a critical component of the extracellular matrix in muscle tissues, providing structural support to muscles and enhancing their strength and resistance. De Paz-Lugo et al. (2018) report that high glycine concentrations (≥1.5 mM, five times the physiological concentrations in human plasma) increase the synthesis of type 2 collagen by 2.5 times in cultured bovine chondrocytes [40]. This cellular collagen increase suggests its potential use for muscle repair after injury or strenuous exercise.

Glycine and high-intensity exercise have been implicated in the upregulation of mTORC1 (mammalian target of rapamycin) signaling, a pathway crucial for muscle protein synthesis [41,42,43]. This upregulation is essential for maintaining muscle mass and promoting hypertrophy, particularly in response to resistance training. Glycine and arginine, through the action of arginine glycine amidinotransferase, increase creatine synthesis in skeletal muscle, a vital molecule for energy production, especially during high-intensity exercise [23]. Moreover, glycine decreases the production of pro-inflammatory cytokines such as TNF-alpha and IL-6 and inhibits the activation of the nuclear factor kappa B (NF-κB) pathway [44]; therefore, it inhibits inflammation and oxidative stress, improving recovery post-exercise. In the context of muscle fatigue, glycine functions as an inhibitory neurotransmitter, playing a role in central fatigue [30], a mechanism contributing to the overall sensation of muscular fatigue during exercise. It may delay muscle fatigue, although the exact processes in this context require further exploration. Although the above studies show that glycine intake could preserve muscle function, protect against muscle wasting, and enhance muscle growth and regeneration, more research and controlled studies on physical performance in athletes are needed because human studies are scarce.

## 5. The Role of Glycine Supplementation on Muscle Strength, Regeneration, and Growth

Animal studies suggest that glycine as a diet supplement promotes muscle growth and regeneration, enhances muscle function, and protects against muscle wasting since glycine activates anabolic pathways, inhibits proteolytic gene expression, regulates cell death, and exerts anti-inflammatory effects [45,46]. Koopman et al. (2017) conclude that glycine supplementation protects against muscle wasting during high-stress conditions (e.g., cancer cachexia, sepsis, and reduced caloric intake) [47]. Kumar et al. (2021), in a randomized clinical trial, reported that 1.33 mmol/kg/day of glycine and 0.81 mmol/kg/day of N-acetylcysteine for 24 weeks decreased inflammation and endothelial dysfunction, increased strength and exercise capacity, and lowered body fat and waist circumference in older and healthy adults (71–80 years). These benefits declined after stopping supplementation for 12 weeks [33]. However, Buchman et al. (1999), in a randomized clinical study on marathon runners, found that injuries to tissue muscle that occurred after completion of the marathon and approximately 48 h later were similar between athletes with or without glycine supplementation (10 g, three times a day for 14 days before the run) [11]. 

Caldow et al. [42] and Sun et al. [46], in in vitro studies of dystrophy and muscle wasting in C2C12 cells, have shown that the administration of glycine protects against muscle wasting and atrophy by activating the mTORC1 complex and inhibiting MuRF1 and Atrogin-1 gene expression. In addition, amino acid supplementation coupled with endurance or resistance training increases mRNA expression in skeletal muscle of several amino acid transporters, including L-type amino acid transporter 1 (LAT1), CD98, sodium-coupled neutral amino acid transporter 2 (SNAT2), and cationic amino acid transporter 1 (CAT1), in both young and older adults [48]. These mechanisms indicate a possible effect of glycine intake and exercise on the synthesis of GlyTs and its beneficial effects on health and physical [31] performance. Although the abovementioned studies indicate that glycine regulates protein turnover and improves strength, muscle regeneration, and growth, controlled clinical trials in humans are necessary to confirm these results.

## 6. Glycine’s Effect on Sleep Quality and Recovery

Beyond its direct effects on muscle function, glycine’s role in promoting sleep quality and recovery presents another contributor to performance optimization. Human studies found that 3 g of glycine before bedtime improves sleep quality, daytime sleepiness, and fatigue induced by acute sleep restriction in healthy men and persons with difficulty sleeping [49,50]. These effects are related to peripheral vasodilation and modulation of sleep-regulating neurons [51,52]. Animal studies report that glycine induces hyperpolarization and cessation of the firing of orexin neurons, increases non-rapid eye movement [52], and proposes that it promotes sleep via peripheral vasodilatation through the activation of N-methyl-D-aspartate (NMDA) receptors in the suprachiasmatic nucleus shell [51]. As sleep is increasingly recognized as a critical factor in athletic recovery and performance [53], exploring glycine’s potential in this context could yield valuable insights for athletes and coaches [49,50]. These effects could indirectly support fitness goals by enhancing recovery processes in high-intensity sports.

## 7. Conclusions

This short review highlights individuals’ deficient attention and care regarding glycine intake as a nutritional supplement since there are no rigorous clinical studies on its nutritional or ergogenic safety and efficacy in humans or its systematic use in the sports environment. The findings are summarized below.

Glycine exhibits cytoprotective effects in several body organs and tissues by modulating oxidative stress, inflammation, and apoptosis, stimulates muscle protein synthesis, is involved in collagen synthesis and sleep regulation, and facilitates muscle repair. Glycine supplementation has been linked to increased resistance, strength, muscle regeneration, growth, and decreased fatigue post-exercise. However, most of these effects have been studied in animal models, isolated cells, and tissues, and, to date,, there is not enough evidence of its usefulness as a sports nutritional supplement with therapeutic or ergogenic properties in athletes, nor to establish dietary recommendations for athletic performance. In addition, it is essential to consider that high doses of glycine (>500 mg/kg of body mass) can induce renal and hepatic cytotoxic effects and contribute to acute toxicity in the brain, causing neuronal death [54,55,56]. Therefore, this review calls on the scientific community to prioritize well-designed, randomized, controlled trials investigating the effects of glycine supplementation on physical performance, muscle strength, recovery, and sleep quality in the human population. In addition, molecular studies incorporating glycine supplementation with physical training, physical fitness measurements, and biochemical analysis of blood samples, adipose, and muscle tissue are needed to corroborate this supplement’s absence of cytotoxic effects in vivo models, as well as its therapeutic effectiveness and possible ergogenic impact.

## 8. Limitations

An extensive search was previously conducted for manuscripts on the effects of glycine supplements on athletes’ physical performance, muscle strength, and skeletal muscle. However, besides the scarcity of manuscripts on human models, the different methodologies did not allow quantitative results to be used. In addition, it was observed that, in general, glycine is not used in isolation but instead combined with other supplements and amino acids, preventing the systematization of the information.

## Figures and Tables

**Figure 1 sports-12-00265-f001:**
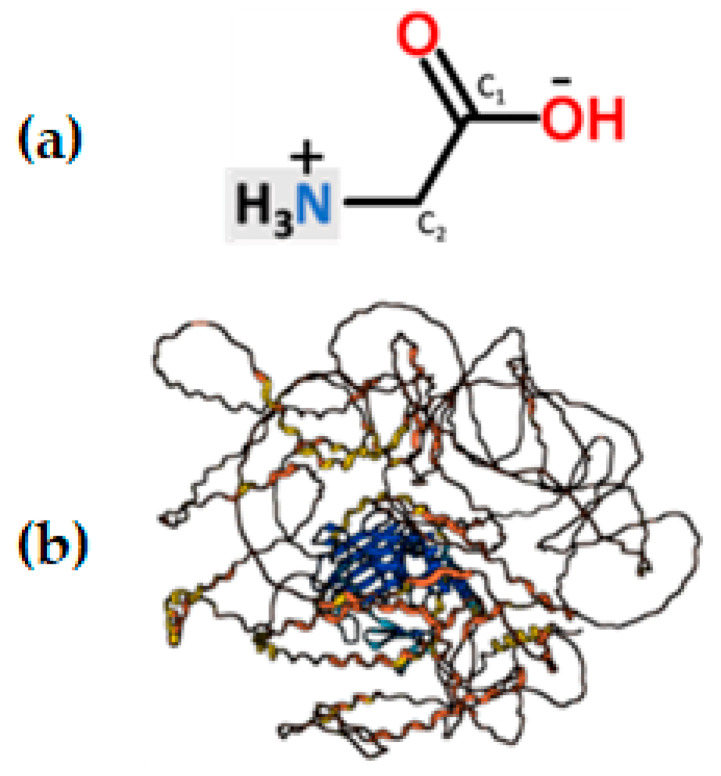
(**a**) Molecular structure of glycine. (**b**) Predicted structure of the alpha−1 collagen protein (COL1A1 human gene; pLDDT score 53.15) where glycine (alpha helix sites) is abundant. Figure generated by AlphaFold 3.0.

**Table 1 sports-12-00265-t001:** The role of glycine in skeletal muscle metabolism.

References	Metabolism Aspect	Role of Glycine
[13,23]	Muscle Protein Synthesis	Stimulates muscle protein synthesis, essential for muscle growth and maintenance.
[23]	Cytoprotecting and tissue repair	Facilitates muscle repair and recovery post-exercise or injury.
[33]	Anti-Inflammatory Effects	Reduces inflammation, beneficial in conditions like arthritis and muscle soreness.

## Data Availability

Any researcher that contacts the corresponding author, M.A.H.-L. (marco.antonio.hernandez.lepe@uabc.edu.mx), will have access to the study data required.

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
