# Peer review of "An Update of the Promise of Glycine Supplementation for Enhancing Physical Performance and Recovery"

_sports, 2024, doi:10.3390/sports12100265_

Round 1
Reviewer 1 Report
Comments and Suggestions for Authors
Background
- Provides good contextualisation
Metabolism and Biochemistry
- Good and concise explanation
- I would link it with the following section
The role of Glycine in Skeletal Muscle Metabolism
- More needs to be said about the processes that involve glycine in recovery and protein synthesis. It is important to highlight what processes occur for it to act in skeletal muscle. In the previous section only fats were discussed.
The role of Glycine on Muscle Strength, Regeneration, and Growth
- Please expand this section further, discussing the theories of studies that have found beneficial effects.
Glycine's Effect in Sleep Quality and Recovery
- Same as previous section.
Reviewer 2 Report
Comments and Suggestions for Authors
I think this is an interesting topic, but I have some comments/critiques. Some are basic (grammar, etc.); some are content related
L36 ...proteins (e.g., collagen in human.....
L40: ref are needed. When you put specific info (in this case synthesis rate, etc) there should be a citation
L46-L52: ref are needed. When you info, like what foods provide glycine - it should be cited
Glycine should be lowercased throughout; except when starting a sentence. There are many instances where it should be lowered
L78: this shouldn't be here. You should have a discussion section where you address limitations and what future studies are needed
You need a purpose statement. The purpose of this review is to..... (right now its not clear as to the purpose/role of the paper)
Also, it would help us to understand your search strategy. Why? When I enter into Pubmed glycine supplementation and human there are 902 results: https://pubmed.ncbi.nlm.nih.gov/?term=glycine+supplementation+AND+human
Your study has 30 references, so it becomes hard to see this as a comprehensive review
I would have section II to be a Methods section detailing your search strategy and presenting your keyword combinations. These should then reflect the purpose of the review
What is the purpose of section II? Your paper seems to be a review of current knowledge metabolism. How does this answer your related to performance and recovery? I am not sure that it does
Sections III-V: is that all of the available research? Seems sparse
L152: don't use "we"
Recommend having a discussion section where you talk about limitations in the research and the need for RCT
Reviewer 3 Report
Comments and Suggestions for Authors
- The topic of glycine supplementation and its potential benefits in physical performance and recovery is timely and relevant. Given the increasing interest in nutritional supplements among athletes and the general population, this review addresses an important area of sports medicine and nutrition.
- The article provides a thorough review of existing literature on glycine, covering a wide range of its physiological roles and potential benefits. The discussion spans multiple aspects of muscle metabolism, strength, regeneration, and sleep, which are all pertinent to athletic performance.
- The authors clearly identify the lack of robust human studies as a major gap in the current research. This is crucial for guiding future investigations and emphasizes the need for more randomized controlled trials to validate the findings from preclinical studies.
- The article presents a balanced view, acknowledging both the promising aspects of glycine supplementation and the limitations of current evidence, particularly the potential cytotoxic effects at high doses. This balanced approach is necessary for a scientific discussion.
However,
- The article is a review and does not present any original research data. While reviews are valuable, the lack of novel findings might limit the impact of the paper in high-impact journals that often prioritize original research. To enhance its value, the authors could consider including a meta-analysis or a systematic review of existing studies to provide more quantitative insights.
- Although the article acknowledges the scarcity of human studies, this limitation significantly weakens the case for glycine as an ergogenic aid. High-quality journals might expect stronger evidence, particularly from human trials, to substantiate the claims made. The authors could have strengthened their discussion by proposing specific research designs or mechanisms by which future studies could overcome current limitations.
- The article relies heavily on preclinical studies (animal and in vitro) to support the potential benefits of glycine. While these studies are important, the translation of these findings to human physiology and performance is not straightforward. The review might benefit from a more critical assessment of the challenges in translating preclinical findings to clinical practice.
- The article's structure could be more streamlined. The transition between sections, particularly between the biochemical roles of glycine and its effects on muscle performance, could be smoother. Additionally, the repetition of certain points (e.g., the role of glycine in muscle metabolism) could be minimized to improve clarity and focus.
- Lacks depth in discussing the implications of the findings and recommendations for future research. A more detailed conclusion that integrates the various aspects discussed in the review and offers a clearer roadmap for future studies would strengthen the manuscript.
- There are minor editorial issues, such as formatting inconsistencies and occasional awkward phrasing, which could be polished to improve readability. Additionally, the figures provided in the article, while relevant, could benefit from more detailed explanations and higher quality visual representation.
Recommendations for Improvement:
- To increase the article's impact, the authors could perform a meta-analysis or systematic review of the existing human studies on glycine supplementation. This would provide a more robust statistical foundation for their claims.
- The review could be enhanced by offering detailed proposals for future research, including specific study designs, target populations, dosage considerations, and potential biomarkers for monitoring glycine's effects.
- The conclusion should be expanded to discuss the broader implications of glycine supplementation in sports nutrition, potential risks, and the importance of personalized approaches to supplementation.
- A thorough editorial review to correct minor errors, improve the flow of the text, and enhance the clarity of figures would be beneficial.
The article has potential for publication, but it may require significant revisions to meet the standards of a high-impact scientific journal. The topic is highly relevant, and the review is comprehensive, but the manuscript could be strengthened by addressing the weaknesses identified above, particularly the need for more rigorous evidence and a more critical analysis of the existing literature. If these revisions are made, the article could contribute valuable insights to the field of sports medicine and nutrition.
Round 2
Reviewer 1 Report
Comments and Suggestions for Authors
Accept in present form
Author Response
Accept in present form
R= The article has been substantially improved thanks to your contributions. We greatly appreciate your support during the process.
Reviewer 2 Report
Comments and Suggestions for Authors
I would like to commend the authors for a stronger paper. The new edits, to me, made the manuscript a better read and addressed prior queries
L87- avoid 1 sentence paragraphs
L87 - is intoxication the right word? When I think of intoxication I think of drunk or having altered states in response to a drug
L157: inhiving? I am not familiar with that word. I Googled it, couldn't find it.
Author Response
I would like to commend the authors for a stronger paper. The new edits, to me, made the manuscript a better read and addressed prior queries
R: We really appreciate your contributions and positive comments about the manuscript.
L87- avoid 1 sentence paragraphs
R: Your observation was considered and the sentence has been joined to the next paragraph that continued with the same idea. Thank you
L87 - is intoxication the right word? When I think of intoxication I think of drunk or having altered states in response to a drug
R: We agree with your observation, so the word has been changed to “The maximum doses of glycine administered without observing altered states or side effects in response to its ingestion (Lines 89-90)”. Thank you
L157: inhiving? I am not familiar with that word. I Googled it, couldn't find it.
R: We appreciate your observation, so the word has been changed with “it inhibits”. Thank you
According to your observations, the article has been improved substantially. We really appreciate all your support during the review process.
Reviewer 3 Report
Comments and Suggestions for Authors
Your article has been much improved but in my opinion it should be published in nutrients rather than sports MDPI journal .
Author Response
Your article has been much improved but in my opinion it should be published in nutrients rather than sports MDPI journal .
R: To increase the visibility of the manuscript we wanted a Special Issue related to Supplementation and Exercise, so a few Special Issues achieve our expectations, including a few of the Journal Nutrients and the chosen one from Sports (Exploring the Role of Acute Supplementation in Exercise Performance). If the Editor considers that there is a better option, of course that we can consider it. Thank you
According to your observations, the article has been improved substantially. We really appreciate all your support during the review process.